# Position: Stop Reactively Patching Your Model Every Time and Start Proactive Test-Driven AI Development

**Nadine Chang** [* 1]   **Maying Shen** [* 1]   **Jialiang Wang** [1]   **Rafid Mahmood** [1 2]   **Jose M. Alvarez** [1]

## Abstract

Many modern AI systems are designed to operate under diverse, open-ended, use-cases. To help generalize deployed systems, many deployed-system maintenance pipelines use a reactive AI flywheel that observes emerging feedback from user behavior (errors) and patches the model accordingly. However, when used as the primary maintenance mechanism, these flywheels often ignore the broader context of these errors within the system's objectives, failing to preempt potential future edge cases, which leads to more unnecessary flywheel iterations. Also, it is statistically increasingly difficult to collect remaining errors due to the long-tail nature of open-world use-cases (Boneh and Hofri, 1997). This position paper argues that a *proactive test-driven flywheel* is required to address reactive flywheel's limitations and to approach a generalizable system. We advocate for creating a "test space" to technically map feedback data to task objectives, evolving the flywheel from reactive to proactive. We augment our position by mathematically proving a proactive one achieves better long-term scaling with fewer iterations than the reactive flywheel.

## 1. Introduction

Recent machine learning progress has spurred several commercial AI systems—from language models to robotics and autonomous driving (AD)—with the ambitious goal of open-ended real-world operation under an ever-expanding set of use cases. These AI systems need extraordinary generalization capabilities to be able to adapt to unknown situations comprised of new user contexts and interactions. We ob-

serve this need especially in physical AI (e.g., robotics and AD systems), where the real-world test distribution of interactions is vast (Palma, 2025; Dnistran, 2025). From the emergence of ImageNet to recent GPT models, the conventional approach to developing general-purpose AI systems has emphasized large, diverse training datasets as a proven method for achieving generalizability (Deng et al., 2009; Achiam et al., 2023). Now, foundation models (FMs) accelerate the process of training a generalized AI system because systems can be easily built on top of FMs, which are trained on diverse, internet-sized data (Rankin, 2025). Concretely, faster development enables faster system deployment.

Once deployed, any test feedback is an invaluable signal to update the system. Now the question is, how do we effectively leverage the signals and fix the system? A common and publicly documented maintenance mode for deployed AI systems is a "flywheel" loop - a continuous loop that improves the system by leveraging system feedback, ranging from minor edge cases to catastrophic failures (e.g. fast, tight turns to crashes in AD). For brevity, we refer to this feedback spectrum as generic "errors". Because the flywheel reacts to real test feedback as they arise, we refer to this paradigm as a "reactive test-driven" flywheel (RF). In this paradigm, feedback data is logged and triaged, usually with the help of some human intervention. A common data-centric practice is to then retrieve similar data to feedback data from a pre-existing data lake or even collect de novo in order to re-train and validate the model(Clips, 2022), while a model-centric practice trains directly on the feedback data (Yao et al., 2021a). RF remains operationally attractive due to its ability to produce visible short-run progress and alignment with the incident-response workflow of software maintenance (Sillito and Kutomi, 2020).

A reactive flywheel faces three major challenges. First, by considering only similar data to errors or training on errors themselves, a purely reactive flywheel focuses on patching the specific observed errors (Ilharco et al., 2022), the opposite goal of generalizability. Consequently, such purely reactive loops remain defensive and fail to prevent future vulnerabilities. For example, after a 2023 fatal autonomous vehicle (AV) crash involving a parked firetruck with active lights, the lack of reported firetruck crashes suggests the

---

[1]NVIDIA  [2]University of Ottawa. Correspondence to: Nadine Chang <nadinec@nvidia.com>, Maying Shen <mshen@nvidia.com>.

*Proceedings of the 43rd International Conference on Machine Learning*, Seoul, South Korea. PMLR 306, 2026. Copyright 2026 by the author(s).

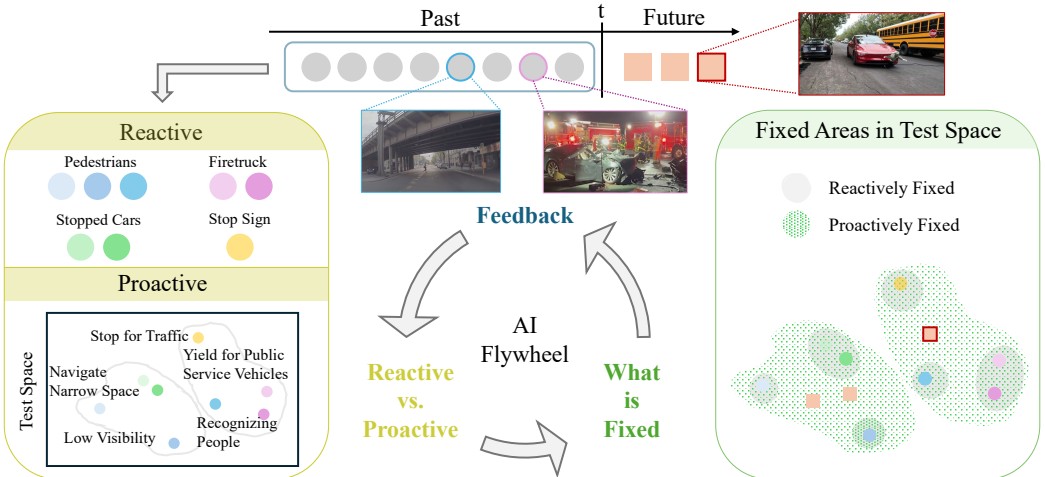

*Figure 1.* Overview of reactive and proactive AI flywheel. In the reactive paradigm, feedback is triaged and specific errors are patched (e.g. 'pedestrian' and 'firetruck'), failing to generalize beyond specific errors. In contrast, the proactive paradigm maps feedback to a test space of required task conditions and addresses the entire task condition (e.g. 'yield for public service vehicles' and 'recognizing people'). This proactivity supports improved generalization and future error prevention. For visual clarity, the test space is illustrated as a 2D space, and each feedback is associated with a single task condition. Because the proactive flywheel structurally organizes feedback differently, the same feedback reactively triaged into one group may fall into different regions of the test space.

specific scenario was likely patched (ABC7, 2023). However, a 2025 potentially fatal crash involving the same AV system and a parked school bus with lights revealed that the root cause—public service vehicles stopped in traffic with pedestrians nearby—remained unaddressed (Dnistran, 2025). Patching the firetruck incident failed to generalize to the functionally identical school bus scenario, as shown in Fig. 1. Secondly, RFs are inefficient. Unable to preempt future errors, RFs lead to more iterations and backlog that strain fixed resources. Lastly, reactive flywheels assume that they will eventually encounter all edge cases. However, as more edge cases are found, it is significantly harder to encounter the remaining cases. This is known as the coupon collector's problem (Boneh and Hofri, 1997) and can be especially dangerous in physical AI, where waiting for extremely rare and potentially unsafe cases to emerge can present unacceptable safety risks, as observed in Koopman (2024).

To overcome these challenges, we advocate that **to achieve generalizable AI systems, reactive test-driven patches are structurally suboptimal, and there is a need for goal-oriented, proactive test-driven flywheel.** Revisiting the automated driving (AD) firetruck and school bus scenarios reveals a common root cause in AD goals: the failure to stop for public service vehicles parked in traffic near pedestrians. National Highway Traffic Safety Administration (NHTSA) (USDOT, 2017) guidebook for AD identifies detecting these vehicles, predicting nearby pedestrians due to association, and responding to these unusual conditions as core requirements. These are also covered in the Department of Motor Vehicles (DMV) handbook (DMV, 2025). If the

initial firetruck error were mapped to the various required driving task conditions, fixing them would have prevented the subsequent school bus incident. We illustrate this example reactive and proactive comparison in Fig. 1. Addressing RF's first limitation and introducing a prevention advantage, a focus on task conditions shifts the flywheel from defensive patching to proactive refinement.

Consequently, we argue that **coverage over a "test space" is needed to evolve a reactive test-driven flywheel to a proactive test-driven flywheel.** Inspired by "train space"—an $n$-dimensional mapping of training data used for data diversity selection in conventional dataset curation (Shen et al., 2025; Diao et al., 2025; Slyman et al., 2024; Sener and Savarese, 2017)— for a given task, we define the "test space" as an $n$-dimensional space encompassing vectorized representations of task conditions without necessitating system data. Operationally, the proactive objective is not merely to name this space, but to improve coverage over its important regions. A concrete atlas-style implementation can view this space as an atlas of task conditions: the atlas defines which conditions can be covered, while a frequency-weighted atlas specifies how strongly each condition should guide updates. In Fig. 1, we visualize the test space's efficacy on the AD example. By mapping system errors into this space, a proactive flywheel can address entire task conditions rather than isolated errors. This setup also encourages automation by bypassing human intervention usually needed for feedback triaging. To augment our position, we mathematically prove even a naive design of our test space proactive flywheel overcomes RF's second limitation by showing significantly better long-term

scaling with fewer flywheel iterations and backlog. Furthermore, because proactive flywheel does not rely on collecting all edge cases to fix potential errors, it lessens the safety consequences of the coupon collector problem, RF's last limitation. Finally, we discuss open challenges to developing a test space-driven proactive flywheel to guide future research.

## 2. Reactive Flywheels in a Nutshell

Because incident-driven and observed-failure-driven update loops are common in deployed ML systems and adjacent research, we discuss first why the reactive AI flywheel remains attractive to the research and industry community (Clips, 2022; Tang et al., 2020; Yao et al., 2021b). We refer to Appendix A for an expanded related research literature driving these motivations.

**Errors are finite and will all eventually emerge.** Reactive error correction implicitly assumes that the universe of possible use-cases, and corresponding errors, is limited. This seems reasonable in closed-set machine learning (ML), which features a finite, enumerable set of tasks (Geng et al., 2020). Reactive iterative strategies are commonplace in ML research, reinforced by concepts from learning curves (Viering and Loog, 2022), hard negative mining (Shrivastava et al., 2016), and active learning (Cohn et al., 1996), which argue that ML models improve over time by addressing discovered weaknesses. This lends the intuition that although the long-tail of edge cases is vast, we are chipping away at it and, crucially, this process will converge fast enough to be practical. The assumption seems reasonable given examples, such as large language models' rapid performance improvement on benchmarks until the benchmark is nearly solved (Brand and Denain, 2025).

**Backlogs of Known Errors Are Temporary.** Reactive maintenance also assumes that any backlog of unfixed errors is a temporary phenomenon, one that engineering teams will whittle down over time. In early deployment, a backlog of issues may accumulate as new error modes pour in, but practitioners typically believe this backlog will shrink as the model matures. This is analogous to the "bug backlog" in software projects, which often peaks right after launch and clears after a few update cycles (Mockus et al., 2002). The ML community carries over this intuition: they expect that after an initial surge of discoveries, the model will reach an equilibrium where new errors are rare and mostly minor, allowing the outstanding issues to be resolved (Sculley et al., 2015; Shrivastava et al., 2016).

## 3. Comparing Reactive and Proactive

Modern AI systems operating in the real world are exposed to an innumerable range of scenarios, spanning countless user contexts and edge cases. However, current reactive flywheels operate under the assumption that these scenarios are finite. To isolate and compare the scaling behavior of reactive and proactive paradigms under a single setting, we introduce a stylized flywheel that favors a reactive setting by following its implicit assumption that errors are finite. **We leave all proofs (of theorems, etc.) in Appendix.**

### 3.1. Model setup

Consider an ML model deployed by a developer. There exists a set of $M$ undiscovered usage scenarios, which the model was not sufficiently trained for initially and would fail on during deployment. These scenarios may correspond to long-tail rare events or potential new usages, which were not highlighted during development. For example in AV, a rare usage scenario may be to "stop for public service vehicles parked in traffic near pedestrians." In practice, while high-traffic deployments encounter repeated incidents, we assume that each $M$ scenario is distinct for analytical convenience. Under the test space design, these $M$ scenarios are characterized by $K$ factors. To further simplify the analysis, we assume that each scenario is associated with a single factor. The factors partition the scenarios evenly, yielding a group of $M/K$ scenarios per factor. These assumptions are made solely for analytical clarity and do not affect the qualitative analysis and comparison between reactive and proactive paradigms. While we define a finite $M$ and $K$, our analysis will focus on the asymptotics of these parameters. We also note that the human workload scales with incident volume, not only with the number of distinct scenarios $M$. The analysis therefore underestimates human attention consumption.

At time $t = 0$, the model is deployed to users. Each time step $t$ reflects a round of an AI flywheel, where the model is used, feedback is given, and the model is updated to address these feedback. *The developer's objective is to discover and update the model to correct all $M$ scenarios within as few flywheel iterations as possible.* In each time step $t$, let $i \in [M]$ be a random scenario drawn uniformly from $M$ scenarios, let $\mathcal{K}(i) \subset [M]$ be the group this scenario belongs to, and let $\mathcal{U}_t$ be the set of scenarios that the model is unable to handle at time $t$, initialized to $\mathcal{U}_0 = [M]$. If $i \in \mathcal{U}_t$, then user interaction results in a model error. The developer logs this event and attempts to update the model to resolve the error. If the developer is successful, then $i$ is removed from $\mathcal{U}_{t+1}$ for the next time step.

Each flywheel iteration incurs costs from incident analysis, data acquisition, retraining, and redeployment. Given a fixed resource budget, a finite number of errors is observed, and the developer can apply different policies for addressing errors. We note that, in practice, developers are resolving a large number of errors per iteration, especially for high-

traffic products. For tractability, we model the one-failure setting, and our key results remain the same when factoring multiple errors. Below, we define two general strategies, one that is reactive and one that is proactive:

1. **Reactive Flywheel:** If $i_t \in \mathcal{U}_t$, then analyze the logged error and update the model with additional data or new training paradigm. Let $p_R \in (0, 1]$ be the probability that the developer resolves the error. Then,

$$\mathcal{U}_{t+1} = \begin{cases} \mathcal{U}_t \setminus \{i_t\} & \text{w.p. } p_R \\ \mathcal{U}_t & \text{w.p. } 1 - p_R \end{cases}$$

2. **Proactive Flywheel:** At each time step, if $i_t \in \mathcal{U}_t$, then analyze the logged error, determine the corresponding scenario group that the error belongs to, and update the model to try to simultaneously resolve all usage scenarios in the group, i.e., proactively addressing unobserved scenarios that share similar underlying causes. Let $p_P \in (0, 1]$ be the probability that the developer resolves all errors in that factor group. Then,

$$\mathcal{U}_{t+1} = \begin{cases} \mathcal{U}_t \setminus \mathcal{K}(i_t) & \text{w.p. } p_P \\ \mathcal{U}_t & \text{w.p. } 1 - p_P \end{cases}$$

To be conservative in favor of RF, we assume $p_P < p_R$, i.e., proactive is more difficult than reactive. Intuitively, proactive correction requires attempting to address potential future errors in broad factor groups, whereas reactive correction focuses specifically on patching a single scenario. Thus, when proactive policy succeeds with a factor-level fix, it removes a group of related scenarios from the long tail rather than only the observed instance.

This model allows us to analyze the consequences of different flywheel strategies. We will first explore the expected number of flywheel iterations required to identify and correct all $M$ usage scenarios. We will then explore the backlog—i.e., the accumulation of logged errors that developers were unable to resolve and thus de-prioritized in favor of new errors.

We note that alternative models of the flywheel may consider how reactive exposure may eventually enable compositional generalization. For instance, after observing enough rare combinations, the ML model may infer a broader rule that covers unseen recombinations. Our analysis can also capture this possibility by treating such structure as a reduction from scenario-level errors to a smaller number of latent factors. The remaining bottlenecks become waiting to encounter the right rare combinations reactively or to identify the shared factors proactively.

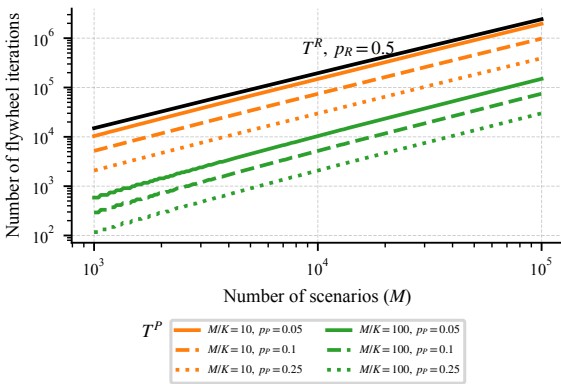

*Figure 2.* The expected number of flywheel iterations required to correct all scenarios under reactive and proactive flywheel (i.e., Thm 3.1 and 3.2). Reactive iterations can grow to multiple orders of magnitude more as the number of scenarios $M$ or the likelihood of addressing an entire factor group $p_P$ increases.

### 3.2. Reactive paradigm needs more flywheel iterations to correct errors

A scenario $i$ is revealed with probability $1/M$ in each flywheel iteration. Under reactive flywheel, the probability of fixing a specific scenario is $p_R/M$. Under proactive flywheel, scenario $i$ is removed whenever any scenario $j$ in $\mathcal{K}(i)$ is exposed, and the factor group-level attempt successfully addresses $j$. If $|\mathcal{K}(i)| = M/K$, the effective per-iteration removal probability of a specific scenario $i$ is $p_P/K$. Given these primitives, we can estimate the expected number of iterations required to correct all errors under both policies.

**Theorem 3.1.** *The expected number of flywheel iterations required to correct all errors under Reactive Flywheel is* $\mathbb{E}[T^R] := (M/p_R) \sum_{s=1}^{M}(1/s)$.

**Theorem 3.2.** *The expected number of flywheel iterations required to correct all errors under Proactive Flywheel is* $\mathbb{E}[T^P] := (K/p_P) \sum_{s=1}^{K}(1/s)$.

The theorems imply that completion time is dominated by the discovery of rare scenarios. The expected time to observe all $M$ scenarios under uniform sampling is a Coupon-collector time (Boneh and Hofri, 1997). Here, $\sum_{s=1}^{M} 1/s = \log M + \gamma + o(1) =: H_M$ is the $M$-th Harmonic number, where $\gamma \approx 0.577$ is the Euler-Mascheroni constant. Consequently, reactive and proactive flywheels scale as $\Theta((M/p_R) \log M)$ and $\Theta((K/p_P) \log K)$, respectively. The trend is shown in Fig. 2, where we can find that reactive paradigm requires more flywheel iterations than proactive and grows in multiple orders of magnitude as the number of scenarios $M$ or proactive success rate $p_P$ increases. Importantly, proactive approach can result in orders of magnitude *fewer* flywheel iterations than the reactive approach, especially when $M/K$ is large, suggesting that many usage scenarios are contextually similar. Even

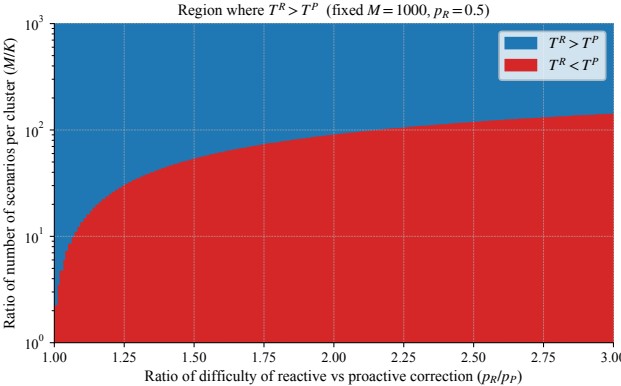

*Figure 3.* Visualizing where proactive flywheel outperforms reactive (in blue) given the difficulty of proactive versus reactive and the number of scenarios per group (i.e., Corollary 3.3). Even if proactive correction is twice as difficult to achieve per iteration, it is attractive if it can address sufficient scenarios. Note log y-axis.

when the task is difficult and $M/K$ is small, suggesting that undiscovered scenarios are dissimilar and spread across different categories, the advantage of the proactive paradigm approaches reactive one.

Below, we formalize the exact conditions where proactive flywheel dominates reactive.

**Corollary 3.3.** *Proactive Flywheel incurs fewer iterations of the flywheel if*

$$\frac{p_P}{p_R} \geq \frac{K \sum_{s=1}^{K} 1/s}{M \sum_{s=1}^{M} 1/s}$$

Corollary 3.3 gives a condition under which proactive flywheel requires fewer iterations even when it is harder per iteration. Here, $p_R/p_P$ quantifies how much "easier" is the task of reactively correcting errors versus a proactive approach. If the number of potential future failures resolved by a proactive strategy, i.e., $M/K$, is sufficiently large with respect to $p_R/p_P$, then proactive correction guarantees fewer flywheel iterations, as shown in Fig. 3. For example, even if reactive patches are two or three times more likely to succeed in a given pass, proactive correction can still require fewer total iterations whenever each successful proactive update covers a sufficiently large factor group. Conversely, when meaningful grouping is weak ($K \approx M$) or proactive correction is orders of magnitude harder, the scaling advantage shrinks toward the reactive baseline. Note that these results are ultimately optimistic by assuming (i) uniform exposure and (ii) no catastrophic forgetting, which uniformly affect both policies.

Importantly, while more flywheel iteration translates directly into more operational cost, the flywheel cycle is not just about compute. It includes incident triage, reproduction, data acquisition, labeling, training, evaluation, and

release review. These steps are largely human-limited, even with highly automated training infrastructure (Sculley et al., 2015). The relevant question is therefore not only "can the model be improved," but also "can the organization keep up with the stream of feedback?"

The earlier results imply that a reactive pipeline spends most of its time chasing the long-tail. Thm. 3.1 shows that even under uniform exposure and no forgetting, reactive completion time is dominated by discovery of rare scenarios. This is the coupon-collector effect. Operationally, this means that the later stages of reactive correction are driven by increasingly infrequent errors that are harder to observe, reproduce, and validate. The work does not become simpler as the system improves. The work becomes more selective and more expensive per marginal improvement.

Finally, we note the importance of reducing the number of AI flywheel iterations due to auxiliary costs. In practice, more frequent retraining creates additional opportunities for regressions, evaluation debt, and coordination overhead. These effects increase the effective cost of a high-retrain regime and further amplify the gap between reactive and proactive flywheel.

### 3.3. Backlogs persist longer and grow larger under the reactive paradigm

When developers fail to address the identified error (i.e. with probability $1 - p_R$ or $1 - p_P$), the scenario becomes a part of the developer's backlog. In the next flywheel iteration, the developer must address the emerging scenario rather than previous ones. Over time, this will result in a growing backlog of scenarios under a fixed resource budget.

Let $\mathcal{O}_t := \{i \in [M] \mid \exists s \leq t, I_s = i, i \in \mathcal{U}_{s-1}\}$ be the set of errors that have been observed up to some point $t$. Then, the number of errors in backlog is $B_t := |\mathcal{O}_t \cap \mathcal{U}_t|$. Below, we quantify the expected size of $B_t$ under the two policies.

**Theorem 3.4.** *Under Reactive Flywheel, the expected backlog at any time $t$ is*

$$\mathbb{E}[B_t^R] = M \left( \left(1 - \frac{p_R}{M}\right)^t - \left(1 - \frac{1}{M}\right)^t \right)$$

**Theorem 3.5.** *Under Proactive Flywheel, the expected backlog at any time $t$ is*

$$\mathbb{E}[B_t^P] = M \left( \left(1 - \frac{p_P}{K}\right)^t - \left(1 - \frac{p_P}{K} - \frac{1 - p_P}{M}\right)^t \right)$$

**Proposition 3.6.** *Assume $p_P/K > p_R/M$. Let*

$$t_0 := \max \left( \frac{\log 2}{\log \frac{1 - p_R/M}{1 - p_P/K}}, \frac{\log 2}{\log \frac{1 - p_R/M}{1 - 1/M}} \right)$$

*Then, for any $t > t_0$, we have $\mathbb{E}[B_t^R] \geq \mathbb{E}[B_t^P]$.*

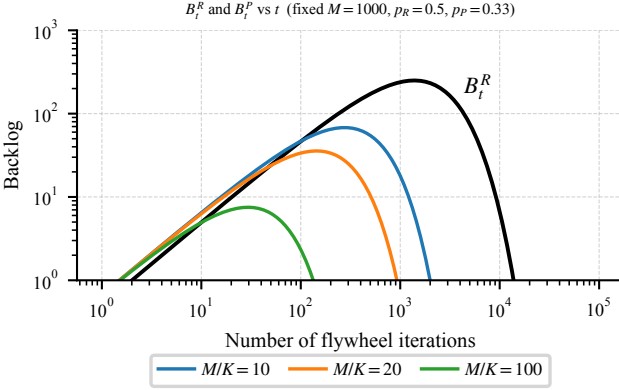

*Figure 4.* Expected backlog after a certain number of flywheel iterations (i.e., Thm. 3.4 and 3.5). Although proactive flywheel may incur slightly larger backlogs in the early stages, proactive backlog falls rapidly, and the maximum backlog is orders of magnitude fewer than that of reactive.

The backlog of any flywheel policy is composed of the gap between how quickly new failure scenarios are discovered (e.g., $(1 - 1/M)$ for reactive policies) and how quickly failures are resolved (e.g., $(1 - p_R/M)$ for reactive policies). Early in deployment, discovery outpaces flywheel, so the gap grows and backlog increases. Later, as more scenarios are discovered and the model usage is better understood, the flywheel dominates. Proposition 3.6 formalizes when the proactive regime overtakes the reactive regime after a finite crossover time.

The trend in backlog accumulation is shown in Fig. 4 and illustrates how the backlog grows early and only decays after discovery saturates. In a reactive regime, backlog therefore persists for a long prefix of deployment time. Note that backlog is not just a list. Backlog creates repeated human obligations. Each unresolved known error must be tracked, prioritized, reassessed across releases, and revalidated when touched by retraining. This is a queueing problem with coordination overhead layered on top, which makes the process more complicated in practice.

From Thm. 3.4 and 3.5 and displayed in Fig. 4, we observe that in the best case—where $K = 1$ implies all scenarios are characterized by a single factor—the proactive paradigm exhibits substantially smaller maximum backlog and resolves backlog significantly faster than the reactive paradigm. In the worst case—where $K = M$ implies a poorly designed test space, in which each factor can only be represented by a single data point—proactive improvement degenerates toward reactive behavior. Even in this extreme setting, the proactive backlog remains no worse than that of reactive.

## 4. How to Build a Proactive Test-Driven Flywheel Using the Test Space

Simply reacting to observed errors can lead to costly flywheel iterations and backlogs. In contrast, proactively addressing errors can reduce these bottlenecks, even if proactive flywheel is per-instance more difficult than the reactive approach. We now discuss how to design a proactive approach using test-driven principles, centered on coverage over the test space. Within this approach, we propose several open research challenges.

### 4.1. Test Space Overview

Proactively improving a system requires mapping observed errors to their underlying usage contexts and designing strategies to improve performance across those underlying domains. For example in AD, user disengagement feedback may show an inability to stop for a firetruck with flashing lights and parked across several highway lanes at night. We can decompose such an observed error into basic factors, such as "yield for public service cars", "caution in the face of flashing lights", or "driving in low visibility". Targeting the underlying factors can correspondingly improve the observed errors and mitigate potential future ones.

We define the space of underlying factors governing a task as a **test space**, i.e., the space of specific testing conditions where the system needs to excel. Given the test space as a primitive, our goal is to map observed errors to mark locations in the test space and build coverage over the important areas of this space to improve the broader task conditions these locations represent. However, developing a test space-driven flywheel requires answering two fundamental questions:

1. How do we create a test space from known testing conditions of the target application?

2. How do we align gathered feedback with the test space to identify uncovered or weakly covered factors driving system evaluation?

Fig. 5 summarizes a coverage-oriented proactive flywheel. In the following subsections, we discuss how to address these two fundamental questions as well as other open problems that need to be addressed for a proactive flywheel.

### 4.2. Creating a Test Space

Rather than emerging from actual data, the test space is envisioned as a structured representation of testing requirements and conditions, as shown in Fig. 5(a). By making the requirements explicit, the test space serves as an anchor for understanding system feedback and provides a reference frame where future feedback can be inferred.

To achieve the test space, it is necessary to first extract

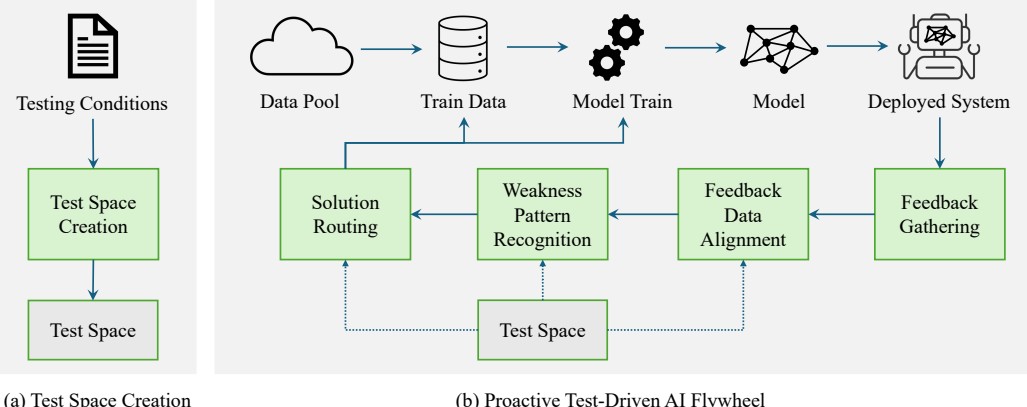

(a) Test Space Creation        (b) Proactive Test-Driven AI Flywheel

*Figure 5.* The paradigm for building a proactive test-driven flywheel. (a) We first create a test space from the task testing conditions, without using any real data. (b) The test space enables a proactive test-driven flywheel. During the loop, gathered feedback is aligned to the test space. Within the test space, weakness patterns associated with the task testing conditions are recognized and later resolved.

concrete factors from existing testing conditions. As decompositions of high-level requirements, these intermediate factors provide a concrete framework for analyzing system feedback. In many physical AI applications, such as AD (USDOT, 2017), the testing conditions are specified in natural language, design documents, or safety requirements. As the system interacts with the real-world, obtaining all possible testing scenarios for factor analysis is costly. Instead, we argue that these written documents provide a scalable, expressive interface for articulating and reasoning through testing conditions and their associated factors.

Fortunately, powerful language models can be helpful to surface and organize the factors implicit in testing requirements. For example, prior natural language processing work (Niklaus et al., 2018) suggests that task-relevant knowledge can be retrieved from textual descriptions; with their broad knowledge, large language models (LLMs) (Naveed et al., 2025) also serve as similarly powerful tools. In practical settings, testing conditions may be accompanied by a few visual examples. In such cases, perception models and vision language models (VLMs) can be leveraged to extract additional knowledge from visual inputs.

> **Open Problem 1.** How can we discover all the concrete factors implicit in the task testing conditions?

Once we identify the set of factors required for successfully achieving the task, we must find a way to represent the space to hold all the factors. We argue that a mathematical and computational representation of the test space is essential for applying ML to analyze these factors and their relationships.

Several prior works represent the space through an embedding space that serves as a proxy abstraction. A variety of these representations includes tokens, network layers from different architectures, or more traditional ML features (Bengio et al., 2013). Such embeddings can be learned

by different methods, such as supervised and contrastive learning (Chen et al., 2020). Alternatively, graph-based abstractions capture factor relationships by modeling dependencies and interactions, complementing embeddings with exposed relational patterns (Hamilton, 2020). Importantly, concepts from train space constructed from real data, such as data coverage, diversity, and similarity, can naturally extend to the test space (Feldman, 2019). Different representation choices—discrete, continuous, or hybrid—are all compatible and remain open research questions.

> **Open Problem 2.** How can we mathematically represent the contextual factors and their relationships that comprise the test space?

Overall, we advocate for the test space as a conceptual layer that decomposes high-level testing conditions into essential factors and as a space where gathered feedback can be mapped to testing requirements.

### 4.3. Developing a Proactive, Test-Driven AI Flywheel

We now explore the components of a proactive AI flywheel that uses the created test space, as shown in Fig. 5(b).

**Feedback Gathering.** Proactive improvement requires access to deployed system feedback, yet feedback gathering alone does not confer proactivity. As seen in Fig. 5(b), feedback is gathered during system execution in real-world conditions and may take different forms, including performance metrics, safety violations, user interactions, or incident occurrence. In many systems, diverse feedback is gathered and filtered in an ad hoc or manual manner. Because feedback can be sparse, such methods limit the gathering scale and timelines (Sculley et al., 2015; Waymo LLC, 2020). We argue that developing methods for automatically identifying and prioritizing informative feedback is a promising research direction. This would reduce human intervention

and enable an efficient, automatic proactive flywheel.

> **Open Problem 3.** How can we automatically identify and prioritize relevant feedback signals for model updating?

**Feedback Data Alignment.** Raw feedback signals are not directly interpretable by the system and thus not actionable for proactive improvement. If feedback is not interpreted in the context of the test space, they remain isolated observations, which primarily support a reactive flywheel. To connect feedback to testing expectations, we must convert feedback into a representation that is compatible with its essential factors and thus the test space. To simplify the problem and, more importantly, to ensure feedback falls into the same latent space, we can break down feedback alignment into two steps. First, we find a method to extract all core factors from a feedback point. Next, these extracted factors are mapped to the same representation used for test space creation, enabling direct alignment with the test space.

In the simplified setting, the main question is what method to use for inferring and extrapolating the factors implicit in a feedback point. For vision data, works on visual explanations and attention mechanisms, such as (Selvaraju et al., 2017; Vaswani et al., 2017), offer methods to identify important regions in an image or video related to specific factors. Beyond vision, extensive studies on language causality (Pearl and Mackenzie, 2018) and counterfactual reasoning (Kaushik et al., 2019) also offer methods using specific words and structures to infer an event, action, or relationships. More broadly, recent advances in multimodal large language models (MLLMs) (Liu et al., 2023) and agentic AI (Yao et al., 2022) highlight increasingly strong multimodal reasoning capabilities, useful for inferring and reasoning through the desired factors.

> **Open Problem 4.** For a given data point, how can we understand all its encompassed factors, which are instantiated within the test space?

**Weakness Pattern Recognition.** Computationally formulating the test space and feedback alignment allows us to identify underlying system weakness patterns from a set of feedback points and proactively address the weaknesses. We can leverage widely studied pattern recognition methods in ML and deep learning. One type of method uses clustering-based approaches to discover structure without explicit supervision. Such methods include k-means, hierarchical clustering, density-based clustering, and distribution-based clustering (Jain, 2010). Other approaches rely on learning-based methods, in which patterns are captured through trained models (e.g. via a support vector machine (Vapnik, 1999) or a neural network (LeCun et al., 2015)).

> **Open Problem 5.** How can we discover system weakness patterns in the test space with mapped data as guidance?

Because weakness patterns are found within the test space—which encodes all testing requirements for a successful task—the identified patterns capture not only the observed scenarios exposed by the collected feedback, but also potential unseen scenarios sharing commonalities. In contrast, a reactive system that addresses only a list of feedback may summarize and address them in a batch, yet lacks the test space as context to generalize beyond them. Consequently, the number of unseen errors that can be anticipated and mitigated is greatly reduced. In the worst case, a poorly designed test space fails to provide additional generalization guidance, and the resulting proactive flywheel's behavior degenerates to that of a reactive system. Therefore, we advocate addressing these patterns generically rather than patching for collected feedback specifically. By doing so, the system can proactively mitigate future feedback and achieve generalization more efficiently.

**Solution Routing.** Lastly, we need to determine how to actually address the recognized patterns, which can be resolved with model-centric and data-centric approaches. From given data, model-centric methods improve model learning by iterating on model architecture, hyperparameters, or training algorithms (Goodfellow et al., 2016; Vaswani et al., 2017). In contrast, data-centric methods improve performance by finding the right data and improving data quality (Ng et al., 2021; Jakubik et al., 2024).

While both paradigms have been extensively studied, they are typically explored in isolation. Most works assume the source of poor performance is known, whether model or data-driven, and propose solutions under this assumption. Diagnosing whether a model weakness is due to a model learning limitation or a data deficiency is understudied.

> **Open Problem 6.** How can we decide what strategy to use to robustly mitigate weakness patterns (e.g., a model- or data-centric manner)?

Conceptually, one natural diagnostic method is to map training data to the test space in a similar manner during feedback alignment. By examining if weakness patterns correspond to sparse training data coverage or densely populated but poorly learned regions in the test space, we can determine if the solution may require more data, better model learning, or both. However, developing a principled diagnosis remains an open direction for proactive AI systems.

## 5. Alternative Views

**Data-rich and bounded tasks.** Strong train and test datasets may already be sufficient for bounded tasks such as closed-set detection. This view is attractive because mature dataset-curation workflows can directly improve coverage, diversity, and benchmark performance without requiring a separate proactive flywheel. However, this approach still depends on knowing, sampling, or constructing the relevant task conditions; when important conditions are rare, shift-

ing, or absent from existing data, dataset coverage alone can leave the same long-tail gap.

**Reactive correction as incident response.** Reactive correction is often a practical engineering choice, because it is fast to implement, has low local engineering cost, fits incident-response workflows, and produces visible short-run progress. These benefits are important for urgent failure events where the AI product needs to be patched with a targeted fix immediately. However, the advantages of reactive correction are local. For instance, if failures are long-tailed or share deeper structure, repeatedly patching observed incidents can still lead to the scaling and backlog costs in Sec. 3.

**When reactive correction may be sufficient.** Our theory also identifies regimes where reactive correction may be viable: the error space may be limited and enumerable, failures may not group into meaningful shared factors ($K \approx M$), or proactive correction may be orders of magnitude harder than reactive correction (very large $p_R/p_P$). Vague and shifting objectives provide another difficult case. For example, generative AI applications that target "virality", such as meme generation, depend on changing social trends, user demographics, and recommendation algorithms (Rathje and Van Bavel, 2025; Berger and Milkman, 2012). In such settings, user feedback may be the most reliable signal, but if the feedback space remains large, the reactive approach still faces the same long-tail cost pressure.

## 6. Conclusion

As AI systems integrate into society, they must achieve the generalizability required to navigate diverse, open-ended use cases. In this position paper, we have argued that transitioning from reactive patching to a proactive, test-driven flywheel is essential for generalizability. Key to this transition is coverage over a "test space" that contextualizes errors within system objectives. Furthermore, we have proved that this proactive flywheel overcomes current reactive limitations by preempting future errors and reducing flywheel iterations. To realize the full potential of a proactive flywheel, we urge AI researchers to help solve the various open problems we have highlighted: test space creation, representation, and coverage, feedback identification and prioritization, feedback alignment with test space, weakness patterns discovery, and weakness pattern resolution. Through interdisciplinary collaboration, we can advance the development of truly generalizable AI.

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

# A. Related Work on AI Flywheel

**Hard Negative Mining.** Many ML systems improve by actively mining "hard" examples that the model gets wrong (false negatives/positives) and retraining on them. Online hard example mining automatically selects difficult cases, yielding faster convergence and higher accuracy (Shrivastava et al., 2016; Lin et al., 2017; Cui et al., 2019; Kalantidis et al., 2020). For object detection in AVs, mining rare object examples (by rarity in feature space) significantly boosts detection of infrequent scenarios (Jiang et al., 2022). Public industry descriptions of autonomous-driving data engines similarly emphasize mining reported or discovered failure cases, retrieving related data from a fleet-scale data source, and adding those slices back into the training set with human or auto-labels (Tesla, 2019; Clips, 2022). This practice is attractive because each mined slice of data produces an immediate, legible improvement in the next model release.

**Red-Teaming and Adversarial Testing.** Before and after deployment, teams conduct structured "red teaming" to probe models for vulnerabilities or harmful behaviors (Ganguli et al., 2022; Bullwinkel et al., 2025). In NLP, this means generating adversarial inputs that induce toxic or unsafe outputs, and then mitigating them (Perez et al., 2022; Zou et al., 2023; Röttger et al., 2024). Organizations invest heavily in adversarial testing; for example, OpenAI's GPT-4 deployment involved over 100 external experts attacking the model to identify issues, followed by model fine-tuning or safety filters as mitigations (Hurst et al., 2024). These public accounts illustrate an observed-failure-driven maintenance mode: teams probe or monitor for failures, add mitigations, and update evaluations after misses. OpenAI's 2025 GPT-4o sycophancy post is a recent example: after deployment signals revealed a behavioral failure, the update was rolled back and the process was revised to add sycophancy-specific evaluations (`https://openai.com/index/expanding-on-sycophancy/`).

**Behavioral Testing.** This is an increasingly popular approach to test set construction that reframes model reliability as a bug-fixing problem (Ribeiro et al., 2020; Srivastava et al., 2023). The key approach involves defining a matrix of capabilities and test types to generate a series of test cases for NLP models. This can be done manually or automatically (Ribeiro and Lundberg, 2022). Such practices assume that comprehensively testing different behavior slices will surface the model's mistakes, which can then be addressed by adding training examples or adjusting the model. Test case generation has also been explored in reinforcement learning planners (Koren et al., 2018).

**Continual Learning.** This framework assumes that ML tasks drift or expand as training data arrives over time (Parisi et al., 2019). The objective is to train a model that preserves it's earlier competence while acclimating to the new settings. This is a fundamental problem as retraining the model repeatedly can risk "catastrophic forgetting", where tasks or settings that were previously learned become recurring failure scenarios for the model. As consumer and general-purpose ML technology grows popular, continual learning has become increasingly prevalent in complex settings including object detection (Shmelkov et al., 2017; Joseph et al., 2021; Singh et al., 2021), robotics (Lesort et al., 2020; Kagaya et al., 2025), and LLMs (Shi et al., 2024).

Taken together, hard-negative mining, active learning, and continual learning primarily address present observed errors or newly arrived data. Behavioral testing is closer to a proactive framing because it constructs behavior slices before every individual failure has been observed, although the usual remediation still updates the model after tests reveal mistakes.

**Incident Response and Patching.** Reactive flywheel is typically implemented as incident response. Systems log failures, route them into ticket queues, and trigger remediation actions to triage the failure, prioritize different failures, retrain the model to resolve the issues, and redeploy the ML model (Xin et al., 2021). Sophisticated ML pipelines in production allow the model to be continuously evaluated, and whenever performance dips or new failure patterns emerge (via user feedback, anomaly detection, etc.), a new training cycle is triggered to deploy a fix (Uber Engineering, 2017; Tesla, 2019). This framework presumes that each known incident can be resolved individually: by pushing a model update or adding post-processing rules, the team eliminates that failure mode. This one-at-a-time servicing keeps the system on track in the short term (and satisfies stakeholders expecting quick fixes), albeit at the cost of growing complexity over time (Sculley et al., 2015).

# B. Proofs

*Proof of Theorem 3.1.* Suppose that at a given time $t$, we have $r := M - |\mathcal{U}_t|$ errors that have been addressed. Then, the probability that we can resolve an additional error at the $t + 1$-th time instance is

$$\Pr(|\mathcal{U}_{t+1}| = M - r - 1 \mid |\mathcal{U}_t| = M - r) = \Pr(I_{t+1} \in \mathcal{U}_t \text{ and the error is fixed} \mid |\mathcal{U}_t| = M - r)$$
$$= \frac{M - r}{M} p_R$$

Furthermore, let $W_r := \min\{\delta \mid |\mathcal{U}_t| = M - r, \mathcal{U}_{t+\delta} = M - r - 1\}$ be the waiting time until the $r + 1$-th error is resolved. Note that this waiting time follows a Geometric distribution based on the Bernoulli probability of resolving an additional error at each time, i.e, $W_r \sim \text{Geometric}((M - r)p_R/M)$. Now, we rewrite $T^R$ in terms of a summation of waiting times

$$\mathbb{E}[T^R] = \sum_{r=0}^{M} \mathbb{E}[W_r] = \sum_{r=1}^{M} \frac{M}{M - r} \frac{1}{p_R} = \frac{M}{p_R} \sum_{r=1}^{M} \frac{1}{M - r} = \frac{M}{p_R} \sum_{s=1}^{M} \frac{1}{s}$$

$\square$

*Proof of Theorem 3.2.* Suppose that at a given time $t$, we have $r := M - |\mathcal{U}_t|$ errors that have been addressed. Then, the probability that we can resolve an additional $M/K$ error at the $t + 1$-th time instance is

$$\Pr(|\mathcal{U}_{t+1}| = M - r - \frac{M}{K} \mid |\mathcal{U}_t| = M - r) = \Pr(I_{t+1} \in \mathcal{U}_t \text{ and the cluster is fixed} \mid |\mathcal{U}_t| = M - r)$$
$$= \frac{K - r}{M} p_P$$

Furthermore, let $W_r := \min\{\delta \mid |\mathcal{U}_t| = M - r, \mathcal{U}_{t+\delta} = M - r - M/K\}$ be the waiting time until the $r + 1$-th error is resolved. Note that this waiting time follows a Geometric distribution based on the Bernoulli probability of resolving an additional error at each time, i.e, $W_r \sim \text{Geometric}((K - r)p_P/M)$. Now, we rewrite $T^P$ in terms of a summation of waiting times

$$\mathbb{E}[T^P] = \sum_{r=0}^{M} \mathbb{E}[W_r] = \sum_{r=1}^{K} \frac{K}{K - r} \frac{1}{p_P} = \frac{K}{p_P} \sum_{r=1}^{K} \frac{1}{K - r} = \frac{K}{p_P} \sum_{s=1}^{K} \frac{1}{s}$$

$\square$

*Proof of Corollary 3.3.* The proof follows from evaluating $T^P > T^R$ and re-arranging the terms. $\square$

*Proof of Theorem 3.4.* For any time $s$, let $A_{i,s} := \{I_s = i\}$ denote the event that the error $i$ was observed at time $s$. Furthermore, let $S_{i,s} := A_{i,s} \cup \{\text{error } i \text{ was fixed}\}$ indicate the event of a successful resolution of the error. Note that $\Pr(A_{i,s}) = 1/M$ and $\Pr(S_{i,s}) = p_R/M$. Finally, define the events

$$E_{i,t} := \bigcup_{s=1}^{t} A_{i,s} \qquad F_{i,t} := \bigcap_{s=1}^{t} \overline{S}_{i,s}$$

denote the events that the error was observed at some point up to time $t$ and that the error was not successfully resolved by time $t$, respectively. Finally, let $X_{i,t} := E_{i,t} \cap F_{i,t}$ denote the event that the error was observed but not fixed by time $t$ and

is currently in backlog. Then,

$$
\begin{aligned}
\mathbb{E}[B_T^R] &= \sum_{i=1}^{M} \mathbb{E}[X_{i,t}] \\
&= M \Pr(X_{1,t} = 1) \\
&= M \Pr(E_{1,t} \cap F_{1,t}) \\
&= M \Pr(F_{1,t}) - \Pr(\overline{E}_{1,t} \cap F_{1,t}) \\
&= M \left( \Pr\left( \bigcap_{s=1}^{t} \overline{S}_{1,s} \right) - \Pr\left( \bigcap_{s=1}^{t} \overline{A}_{1,s} \cap \bigcap_{s=1}^{t} \overline{S}_{1,s} \right) \right) \\
&= M \left( \Pr\left( \bigcap_{s=1}^{t} \overline{S}_{1,s} \right) - \Pr\left( \bigcap_{s=1}^{t} \overline{A}_{1,s} \right) \right) \\
&= M \left( \left(1 - \frac{p_R}{M}\right)^t - \left(1 - \frac{1}{M}\right)^t \right)
\end{aligned}
$$

Above, the first three equalities follow from the definition of $X_{i,t}$, and the fourth equality follows from the definitions of the intersection of two sets. The fifth equality substitutes the definitions of $E_{1,t}$ and $F_{1,t}$ and the sixth equality uses the fact that $S_{1,s} \subseteq A_{1,s}$ for any $s$. Finally, we substitute in the probabilities for $\overline{S}_{1,s}$ and $\overline{A}_{1,s}$. $\qquad\square$

*Proof of Theorem 3.5.* Let $B_{t,1}^C$ denote the backlog corresponding only to the $M/K$ errors in the first cluster. By symmetry of the clusters, $\mathbb{E}[B_t^C] = K\mathbb{E}[B_{t,1}^C]$. Consequently, we will bound $\mathbb{E}[B_{t,1}^C]$.

For each time instance $s \leq t$, let $O_s$ denote the event that the observed error does not belong to the cluster (with probability $1 - 1/K$), let $F_s$ denote the event that the observed error belongs to the cluster but we fail to correct it (with probability $(1 - p_P)/K$), and let $S_s$ denote the success event that the observed error belongs to the cluster and we correct it (with probability $p_P/K$). Finally, let $N_O$, $N_F$, and $N_S$ denote the counts of these events after $t$ observations and note that $(N_O, N_F, N_S) \sim \text{Multinomial}(t, 1 - 1/K, (1 - p_P)/K, p_P/K)$.

We first compute over the $t$ observed errors, the probability that cluster 1 was observed $n$ times and is still uncorrected:

$$
\Pr(N_0 = t - n, N_F = n, N_S = 0) = \binom{t}{n}\left(1 - \frac{1}{K}\right)^{t-n}\left(\frac{1 - p_P}{K}\right)^n.
$$

Conditioned on $n$ observed errors from cluster 1, let $Y_j$ be a binary variable indicating whether error $j \in \{1, 2, \cdots, M/K\}$ was observed and let $D_n = \sum_{j=1}^{M/K} \mathbb{1}\{Y_j = 1\}$ be the count of the number of unique errors from cluster 1 that were observed. Then, conditioned on cluster 1 not being resolved, the expected backlog from the cluster is

$$
\begin{aligned}
\mathbb{E}[D_n] &= \mathbb{E}\left[ \sum_{j=1}^{M/K} \mathbb{1}\{Y_j = 1\} \right] \\
&= \sum_{j=1}^{M/K} \Pr(Y_j = 1) \\
&= \frac{M}{K}\left(1 - \Pr(Y_1 = 0)\right) \\
&= \frac{M}{K}\left(1 - \left(1 - \frac{K}{M}\right)^n\right)
\end{aligned}
$$

Above, the first two equalities follow by definition, the third equality follows from the symmetry of all errors being equivalent, and the fourth equality follows from the fact that $\Pr(Y_1 = 0)$ follows a binomial distribution with success probability $1 - K/M$ and $n$ successes.

Finally, we are ready to bound the unconditional expected backlog from cluster 1:

$$
\begin{aligned}
\mathbb{E}[B_{t,1}^C] &= \sum_{n=0}^{t} \mathbb{E}[D_n] \Pr(N_0 = t - n, N_F = n, N_S = 0) \\
&= \frac{M}{K} \sum_{n=0}^{t} \left(1 - \left(1 - \frac{K}{M}\right)^n\right) \binom{t}{n} \left(1 - \frac{1}{K}\right)^{t-n} \left(\frac{1 - p_P}{K}\right)^n \\
&= \frac{M}{K} \left(\sum_{n=0}^{t} \binom{t}{n} \left(1 - \frac{1}{K}\right)^{t-n} \left(\frac{1 - p_P}{K}\right)^n - \sum_{n=0}^{t} \binom{t}{n} \left(1 - \frac{1}{K}\right)^{t-n} \left(\frac{1 - p_P}{K}\right)^n \left(1 - \frac{K}{M}\right)^n\right) \\
&= \frac{M}{K} \left(\left(1 - \frac{p_P}{K}\right)^t - \left(1 - \frac{p_P}{K} - \frac{1 - p_P}{M}\right)^t\right)
\end{aligned}
$$

Above, the first two equalities follow from definition, the third equality expands the first term in the summation, and the fourth equality applies the Binomial Theorem to both sums to simplify them.

Finally, we substitute $K\mathbb{E}[B_{t,1}^C] = \mathbb{E}[B_t^C]$ to complete the proof. $\qquad \square$

*Proof of Proposition 3.6.* We first note two identities. First,

$$
\frac{p_P}{K} > \frac{p_R}{M} \iff 1 - \frac{p_R}{M} > 1 - \frac{p_P}{K} \iff 1 > \left(\frac{1 - \frac{p_P}{K}}{1 - \frac{p_R}{M}}\right)^t
$$

which holds for any $t > 1$ Let $t_1 := \frac{\log 2}{\log \frac{1 - p_R/M}{1 - p_P/K}}$. Then for any $t > t_1$, we must have

$$
\left(\frac{1 - \frac{p_P}{K}}{1 - \frac{p_R}{M}}\right)^t \leq \frac{1}{2}.
$$

Second,

$$
1 > p_R \iff 1 - \frac{p_R}{M} > 1 - \frac{1}{M} \iff 1 > \left(\frac{1 - \frac{1}{M}}{1 - \frac{p_R}{M}}\right)^t
$$

which holds for any $t > 1$. Let $t_2 := \frac{\log 2}{\log \frac{1 - p_R/M}{1 - 1/M}}$. Then, for any $t > t_2$, we must have

$$
\left(\frac{1 - \frac{1}{M}}{1 - \frac{p_R}{M}}\right)^t \leq \frac{1}{2} \iff \frac{1}{2} = 1 - \frac{1}{2} \leq 1 - \left(\frac{1 - \frac{1}{M}}{1 - \frac{p_R}{M}}\right)^t
$$

We are now ready to prove the inequality. Consider $t > \max(t_1, t_2)$. Then,

$$
\begin{aligned}
\mathbb{E}[B_t^C] &= M \left(\left(1 - \frac{p_P}{K}\right)^t - \left(1 - \frac{p_P}{K} - \frac{1 - p_P}{M}\right)^t\right) \\
&\leq M \left(1 - \frac{p_P}{K}\right)^t \\
&= M \left(1 - \frac{p_R}{M}\right)^t \left(\frac{1 - \frac{p_P}{K}}{1 - \frac{p_R}{M}}\right)^t \\
&\leq \frac{M}{2} \left(1 - \frac{p_R}{M}\right)^t \\
&\leq M \left(1 - \frac{p_R}{M}\right)^t \left(1 - \left(\frac{1 - \frac{1}{M}}{1 - \frac{p_R}{M}}\right)^t\right) \\
&= M \left(\left(1 - \frac{p_R}{M}\right)^t - \left(1 - \frac{1}{M}\right)^t\right) = \mathbb{E}[B_t^B]
\end{aligned}
$$

Above, the first inequality follows from removing the negative term, the second inequality follows from the first identity that we noted earlier, and the third inequality follows from the second identity that we noted earlier. □

