# OpenReview forum: "Position: Stop Reactively Patching Your Model Every Time and Start Proactive Test-Driven AI Development"
_ICML.cc/2026/Position_Paper_Track — ICML 2026 Position Paper Track regular_

### Official Review · Reviewer_Fi4R · 2026-03-12

**Significance:** 3
**Argument Clarity:** 3
**Rating:** 4
**Confidence:** 4

**Questions:**

see weakness

**Alternative Views Section:**

Yes

**Compliance With Llm Reviewing Policy A Conservative:**

Affirmed.

**Discussion Potential:**

3

**Final Justification:**

good paper

**Paper Summary:**

This paper pointout the deficiencies of the reactive test-driven AI flywheel in modern open-world AI systems: It only patches isolated emerging errors, suffers from low iteration efficiency, and is restricted by the coupon collector problem of long-tail edge cases in open scenarios.
To solve these issues, this paper recomand to use proactive test-driven AI flywheel centered on a test space, which maps system errors to task-oriented condition dimensions rather than individual cases to realize proactive error prevention. Through mathematical modeling and theorem proving, the paper try to verifies that proactive test-driven AI flywheel is a better choice.
Furthermore, the paper elaborates on the definition and construction method of the test space, as well as the complete workflow of the proactive flywheel, and puts forward six open research problems for the practical implementation of the framework.

**Position:**

Yes

**Position In Title:**

Yes

**Related Work:**

3

**Strengths And Weaknesses:**

Strengths
- Introduce core position with illustrative figures, which distinguishes the reactive and proactive AI flywheel paradigms for readers.
- Conducts rigorous mathematical modeling and formal theorem proving to validate that the proactive test-driven flywheel is a better choice.
- Proposes a comprehensive set of open research problems, which may provide guidance for future research.

Weaknesses
- The Alternate Viewpoints section is relatively brief and lacks sufficiently strong, contrasting perspectives compared with other position papers in my batch.
- This paper overemphasizes the drawbacks of reactive patching while understating its practical strengths in industry (e.g., fast implementation, low engineering cost). The comparison between two paradigms is unbalanced.

**Support:**

3

---

> ### Author Rebuttal · Authors · 2026-03-30
>
> Thanks for your detailed review and positive overview of our paper.
>
> 1. > The Alternate Viewpoints section is relatively brief and lacks sufficiently strong, contrasting perspectives compared with other position papers in my batch.
>
> Thanks for this suggestion, which is similarly echoed by reviewer `bSSo`. To address this shared concern: Our revision will strengthen the Alternate Viewpoints section by better clarifying the attractive aspects of reactive approaches from a practical perspective, i.e., the difficulty of proactive construction and core limitations. As our theoretical results suggests, reactive correction is viable when the error space is limited, if errors cannot be grouped, and if per-pass reactive correction is orders of magnitude more easily implementable than proactive correction.
>
>
> 2. > This paper overemphasizes the drawbacks of reactive patching while understating its practical strengths in industry (e.g., fast implementation, low engineering cost). The comparison between two paradigms is unbalanced.
>
>
> Reactive approaches offer clear practical benefits, including implementation speed, incident-response fit, and visible short-run progress.
> Our theoretical analysis discusses some of these points mathematically (e.g., if reactive patching is per-pass easier than proactive $p_R >> p_P$, if failures cannot be sufficiently grouped into large categories $K/N \approx 1$). However, we will revise the theoretical discussion and Alternate Viewpoints to more directly explain, practically, when these may make reactive correction attractive.

---

> > ### Author Rebuttal · Reviewer_Fi4R · 2026-04-02
> >
> > none

---

### Official Review · Reviewer_bSSo · 2026-03-12

**Significance:** 3
**Argument Clarity:** 2
**Rating:** 4
**Confidence:** 4

**Questions:**

See weaknesses above.

**Alternative Views Section:**

Yes

**Compliance With Llm Reviewing Policy A Conservative:**

Affirmed.

**Discussion Potential:**

3

**Final Justification:**

Sufficient clarification from authors during the rebuttal.

**Paper Summary:**

This paper addresses approaches for patching models, i.e., the period after deployment, but prior to retraining, to handle bugs and issues of the current deployed models. The authors criticize the conventional reactive approach, which aims to collect similar failures that are encountered during testing, and train a fix but not addressing the root cause and hence leading to worse generalization. Instead they propose a proactive approach similar to root cause analysis that goes beyond superficial similarity and might be better for generalization of mitigations as well as reducing the degree of backlog.

**Position:**

Yes

**Position In Title:**

Yes

**Related Work:**

2

**Strengths And Weaknesses:**

Strengths:
- The paper is fairly well written, makes an intuitive point of reactive vs proactive approaches, and uses a salient motivating example to ground the reader.
- The additional theoretical analysis and simulations contrasting the effectiveness of reactive vs proactive approaches and its backlog is also compelling and strengthens the paper.

Weaknesses:
- While the paper successfully challenges the assumption that the number of errors is limited, it fails to discuss the ease or effectiveness of making reactive patches as opposed to proactive patches. Their theoretical analysis is based on this assumption and there is not discussion or empirical evidence to back this up, therefore, a better head-to-head real-world evaluation of reactive vs proactive patches are needed.
- On a similar note, the paper does not address what additional abilities might be needed to do autonomous proactive patching, such as inductive reasoning, proposing hypothesis, and causal analysis -- it is unclear to the extent to which current models have these capabilities and what are the impacts false negatives or wrong hypothesis which are much less likely in a reactive approach.
- Minor: the alternate view points section can be better structured to explicitly state the main idea, and the arguments for and against it.

**Support:**

3

---

> ### Author Rebuttal · Authors · 2026-03-30
>
> Thanks for your detailed review.
>
>
> 1. > While the paper successfully challenges the assumption that the number of errors is limited, it fails to discuss the ease or effectiveness of making reactive patches as opposed to proactive patches. Their theoretical analysis is based on this assumption and there is not discussion or empirical evidence to back this up, therefore, a better head-to-head real-world evaluation of reactive vs proactive patches are needed.
>
>
>
> We want to clarify that our theoretical model incorporates the reality that proactive patches are harder than reactive patches in any given iteration, via the assumption that $p_P < p_R$ (i.e., the likelihood of a successful patch is lower). However, our theoretical argument shows that even if reactive patches are 2 to 3 times more likely to be successful, the proactive approach will still be better in the long run due to resolving groups of failures. Our position is effectively that reactive patches are only myopically attractive. In our revision, we will make the conservative nature of the setup more explicit in terms of a scaling comparison over time.
>
>
> We agree that a real-world head-to-head is valuable, but developing this proactive pipeline requires answering several key research questions. We envision building a complete flywheel with empirical validation in future work.
>
>
>
> 2. > On a similar note, the paper does not address what additional abilities might be needed to do autonomous proactive patching, such as inductive reasoning, proposing hypothesis, and causal analysis -- it is unclear to the extent to which current models have these capabilities and what are the impacts false negatives or wrong hypothesis which are much less likely in a reactive approach.
>
> We agree that autonomous proactive patching would require capabilities such as factor induction, hypothesis formation, and causal diagnosis, and that the current draft should acknowledge this more explicitly. Our intent is to highlight them as part of the research agenda, which is why the paper emphasizes open problems in factor discovery, feedback alignment, weakness-pattern recognition, and solution routing. We will also make clearer that proactive systems can fail through missed factors, false hypotheses, and poor routing, even if we still believe the proactive framing is the right long-run direction.
>
>
> 3. > Minor: the alternate view points section can be better structured to explicitly state the main idea, and the arguments for and against it.
>
>
> Thanks for this suggestion, which is similarly echoed by reviewer `Fi4R`. To address this shared concern: Our revision will strengthen the Alternate Viewpoints section by better clarifying the attractive aspects of reactive approaches from a practical perspective, i.e., the difficulty of proactive construction and core limitations. As our theoretical results suggests, reactive correction is viable when the error space is limited, if errors cannot be grouped, and if per-pass reactive correction is orders of magnitude more easily implementable than proactive correction.

---

> > ### Author Rebuttal · Reviewer_bSSo · 2026-04-02
> >
> > I think this is a borderline work, but I have increased my score accordingly.

---

### Official Review · Reviewer_NPVH · 2026-03-12

**Significance:** 2
**Argument Clarity:** 3
**Rating:** 4
**Confidence:** 4

**Questions:**

none

**Alternative Views Section:**

Yes

**Compliance With Llm Reviewing Policy A Conservative:**

Affirmed.

**Discussion Potential:**

1

**Final Justification:**

Thanks to the authors for the rebuttal; I acknowledge that even with the opaqueness of how industry deals with problems, and the difficulty of being proactive, there can be some value to serve a reminder.

However, the nature of the topic and position does demand clearer evidence and detailed investigations into the realities and necessities of why reactive approaches continue to be used, if they indeed are to a significant extent, and this is difficult when writing about closed industry practices. I'm not adequately convinced of the relevance of this position and discussion to the typical ICML audience to push my rating up to accept-territory, but I'm improving the score and defer to the other reviewing opinions.

-----
post discussion with AC, bumping up score to 4

**Paper Summary:**

The submission calls for a move away from reactive patching of failures of AI systems to proactive analysis of failure modes which would improve generalization, by better identifying the broader category of failures and then coming up with appropriate measures, which could tackle underlying problems at the modelling level or through data interventions.

**Position:**

Yes

**Position In Title:**

Yes

**Related Work:**

2

**Strengths And Weaknesses:**

Strengths:

The push towards favouring a proactive approach over a reactive one is a sensible one, and is intuitively more generalizable and the correct approach to system failures. The audience for this message is more industry application than research, and might be impactful in real life by pointing in the right direction.

Weaknesses:

It is not clear to me that the submission is justified (or that it justifies itself well) in claiming that reactive patches are universal in practice. Since most of what industry does is hidden from public view, it is not in fact known whether industry does not already do its best to take a proactive approach overall, even if it uses a reactive one sometimes for patching urgent failures. The only examples provided are some anecdotal ones, all involving autonomous driving, and it is not clear to me that a broader proactive approach isn’t taken over long-term horizons anyway by AD companies. The submission claims there is “widespread adoption of the reactive AI flywheel across AD, ranking, search, and LLMs”, but such a sweeping characterization calls for more evidence; we don’t know this practice is indeed so prevalent or not.

The message is quite intuitive, and it seems very unlikely to me that the industry isn’t aware that fixing underlying problems is the better solutions; the problem is that it is not easy to actually identify such factors. The submission makes some suggestions, but once again, we have no way of knowing that such approaches aren’t already in practice.

It’s not fully clear in the context of deep learning systems if the coupon collector problem is the right analogy; couldn’t it be that a certain number of collected edge-cases pushes the model into generalization, because now you have seen the required number of recombinations to enable effective combinatorial generalization?

**Support:**

1

---

> ### Author Rebuttal · Authors · 2026-03-30
>
> Thanks for your detailed review.
>
> 1. > It is not clear to me that the submission is justified (or that it justifies itself well) in claiming that reactive patches are universal in practice. Since most of what industry does is hidden from public view, it is not in fact known whether industry does not already do its best to take a proactive approach overall, even if it uses a reactive one sometimes for patching urgent failures. The only examples provided are some anecdotal ones, all involving autonomous driving, and it is not clear to me that a broader proactive approach isn’t taken over long-term horizons anyway by AD companies. The submission claims there is “widespread adoption of the reactive AI flywheel across AD, ranking, search, and LLMs”, but such a sweeping characterization calls for more evidence; we don’t know this practice is indeed so prevalent or not.
>
>
> Thanks for bringing this important distinction. We do not intend to claim a universal industry behavior, but rather that this reactive patch-and-retrain loop is a common and publicly documented pattern in industry-facing deployed ML systems as well as adjacent research.
> In industry practice, we note two prominent examples: the note on patching sycophancy in OpenAI's GPT-4o [1], and the walkthrough of Tesla's data engine on Tesla's AI Day 2022 [2].
> In related research, addressing the present errors --- as opposed to anticipating potential future errors --- is the prevailing philosophy in continual learning, active learning, and hard-negative mining. Indeed, only more recent research such as in behavioral testing appears to be proactive in nature.
> Our point is therefore not that all practice is purely reactive, but that reactive incident-driven maintenance remains a central operational mode and is insufficient by itself for broad generalization in open-ended settings.
> In our revision, we will reduce universal-sounding phrases, and better revise our related literature to clarify the current research landscape and recent industry examples.
>
> [1] https://openai.com/index/expanding-on-sycophancy/
>
> [2] https://gist.github.com/L0rdCha0s/de22ae0c7e7a7a70b37ac9c1262e27e1
>
> 2. > The message is quite intuitive, and it seems very unlikely to me that the industry isn’t aware that fixing underlying problems is the better solutions; the problem is that it is not easy to actually identify such factors. The submission makes some suggestions, but once again, we have no way of knowing that such approaches aren’t already in practice.
>
>
> We agree that practitioners are not only aware that root-cause-level fixes are desirable, but also aware that it is difficult to find and fix root causes with current techniques. We point to a prominent example: Anthropic's research proving how despite employing standard techinques for explicit safety training to address root causes, LLMs can deceptively hide its ability to avoid said safety training [3]. It is because of this desireability and extreme difficulty that we are calling for more research in several necessary components to realize a proactive flywheel: factor discovery, feedback alignment, and weakness-pattern recognition. To the best of our knowledge, these are open research problems and thus have not been set to practice.
>
>
> [3] Hubinger, Evan, Carson Denison, Jesse Mu, Mike Lambert, Meg Tong, Monte MacDiarmid, Tamera Lanham et al. "Sleeper agents: Training deceptive llms that persist through safety training." arXiv preprint arXiv:2401.05566 (2024).
>
> 3. > It’s not fully clear in the context of deep learning systems if the coupon collector problem is the right analogy; couldn’t it be that a certain number of collected edge-cases pushes the model into generalization, because now you have seen the required number of recombinations to enable effective combinatorial generalization?
>
>
> Thanks for this interesting suggestion. Although we use the coupon-collector analogy as a stylized account, we do think it is appropriate for our regime: when failures are long-tail, costly to discover, and sporadically exposed, the bottleneck is the time required to encounter the right rare scenarios.
> Even under the case of combinatorial generalization, the waiting time to accumulate the right rare combinations can be too slow or too costly in open-world settings.
> We see combinatorial generalization as another potential latent structure (akin to groups of errors) that may be structurally determined. Our revision will include this discussion on such alternate frameworks and how they connect back to the primary bottleneck addressed by proactive correction.

---

> > ### Author Rebuttal · Reviewer_NPVH · 2026-04-02
> >
> > Thanks to the authors for the rebuttal; I acknowledge that even with the opaqueness of how industry deals with problems, and the difficulty of being proactive, there can be some value to serve a reminder.
> >
> > However, the nature of the topic and position does demand clearer evidence and detailed investigations into the realities and necessities of why reactive approaches continue to be used, if they indeed are to a significant extent, and this is difficult when writing about closed industry practices. I'm not adequately convinced of the relevance of this position and discussion to the typical ICML audience to push my rating up to accept-territory, but I'm improving the score and defer to the other reviewing opinions.

---

### Official Review · Reviewer_ngcq · 2026-03-13

**Significance:** 3
**Argument Clarity:** 3
**Rating:** 4
**Confidence:** 3

**Questions:**

N/A

**Alternative Views Section:**

Yes

**Compliance With Llm Reviewing Policy A Conservative:**

Affirmed.

**Discussion Potential:**

3

**Paper Summary:**

This paper argues a proactive test-driven flywheel is required to address reactive flywheel’s limitations and to approach a generalization system. To this end, it proposes a “test space” to technically map feedback data to task objectives. This paper provides solid mathematical support to its claim.

**Position:**

Yes

**Position In Title:**

Yes

**Related Work:**

3

**Strengths And Weaknesses:**

Strengths:
1. Solid mathematical support. The mathematical part is solid and rigorous.
2. The idea of test space is interesting.
3. Potentially,  it has a lot of real-world applications.

Weaknesses:
1. It lacks empirical validation. While this paper is mathematically solid, some empirical studies would make the claim more convincing.
2. It lacks concrete discussion of the test space construction. Although LLMs are suggested as tools for extracting factors, the exact methodology for building a comprehensive and actionable test space remains unclear to me.

**Support:**

2

---

> ### Author Rebuttal · Authors · 2026-03-30
>
> Thanks for your detailed review and positive overview of our paper.
>
>
> 1. > It lacks empirical validation. While this paper is mathematically solid, some empirical studies would make the claim more convincing.
>
>
> We agree that an empirical validation would strengthen the paper, but our current goal is a position contribution and call for research rather than an end-to-end implementation of a proactive flywheel. Our "solid and rigorous" mathematical analysis supports the importance of a proactive flywheel, and we pose several major research questions that are necessary to empirically validate this. We envision an implementation and validation as an important future direction.
>
> 2. > It lacks concrete discussion of the test space construction. Although LLMs are suggested as tools for extracting factors, the exact methodology for building a comprehensive and actionable test space remains unclear to me.
>
>
>
> Thanks for this suggestion. We agree that the current draft can be more concrete about constructing the test space.
> Our intended workflow is:
> 1. Begin from task requirements and safety documents
> 2. Extract the underlying factors they imply
> 3. Represent those factors and their relations in a structured space
> 4. Align observed failures to that space
> 5. Use the resulting patterns to guide model- or data-centric fixes.
>
> We emphasize that LLMs are merely viable tools for extracting factors from written requirements and documents. We will make this pipeline more explicit in the paper, including a short worked version of the AD example, so the proposal reads more operational.

---

> > ### Author Rebuttal · Reviewer_ngcq · 2026-04-04
> >
> > Thanks for the reply and the clarification of test space construction. As I stated in the original review, the theoretical part of this paper is solid. Although I can accept the argument that the empirical studies is not indispensable in this stage, some empirical studies can make the position more convincing. Overall, I think this is a borderline paper and I will maintain my score.

---

### Decision · Program_Chairs · 2026-04-30

**Decision:**

Accept (regular)

**Comment:**

The paper argues for shifting AI development methodologies from reactive patching to proactive, test-driven approaches.
The reviewers found that the position that the authors argue for is well-substantiated, via a solid mathematical framework that the reviewers regarded as a strength of the paper. The paper is well-written and is likely to spark discussion in the field around an important topic. It offers a comprehensive set of open research problems for future work.

The most significant point of contention, raised by Reviewer NPVH, was whether the claims of the paper primarily refer to industry, and in that case (i) without visibility into industry practices, the paper should refrain from making sweeping statements, and (ii) it is unlikely that industry is not aware that proactive development is generally better than reactive patching, since this is intuitive, in which case the position paper is likely to not have impact, so if industry isn't opting for proactive solutions it's likely due to practical difficulties. During the rebuttal, the authors clarify that they don’t intend to make universal claims about industry behaviour, but that reactive patching is a common and publicly documented pattern that they offer references for and additional concrete examples. The authors committed to adjust statements accordingly. The authors also agreed with the reviewer regarding the difficulty in finding root causes, which makes it harder to adopt proactive methods in practice, but they emphasize that this further justifies their call for research in this area. During internal discussions, the reviewer agreed that the paper can serve as a motivation for important research directions in academia along the lines of what the authors suggested in their rebuttal.

Reviewers bSSo also pointed out that the paper fails to recognize that proactive fixes are harder and may require additional knowledge or abilities that aren’t always available. The authors have responded productively to this criticism, noting that this is captured by their mathematical framework, but also committing to strengthen the Alternate Viewpoints section by better clarifying the attractive aspects of reactive approaches from a practical perspective, i.e., the difficulty of proactive construction and core limitations.

The consensus across all reviewers shifted to "Borderline Accept" following the discussions. I therefore recommend acceptance and encourage the authors to make the suggested revisions for the camera ready version.